# Modified Hazelnut Shells as a Novel Adsorbent for the Removal of Nitrate from Wastewater

**Marija Stjepanović** *, **Natalija Velić** and **Mirna Habuda-Stanić**

Faculty of Food Technology Osijek, Josip Juraj Strossmayer University of Osijek, Franje Kuhača 18, 31000 Osijek, Croatia; natalija.velic@ptfos.hr (N.V.); mirna.habuda-stanic@ptfos.hr (M.H.-S.)
* Correspondence: marija.nujic@ptfos.hr

**Abstract:** The aim of the study was to prepare a novel adsorbent by chemical modification of hazelnut shells and evaluate its potential for the nitrate removal from model solutions and real wastewater. The characterization of the novel adsorbent, i.e., modified hazelnut shell (MHS) was performed. The adsorbent characterization included the analysis of elemental composition and the surface characteristics analysis by scanning electron microscopy (SEM) and Fourier transform infrared spectroscopy (FTIR). The adsorption experiments (batch technique) were performed to investigate the effects of adsorbent concentration, contact time, initial nitrate concentration, and solution pH. The nitrate removal efficiency increased with the increase in MHS concentration and decreased with the initial nitrate concentration. MHS was found to be effective in nitrate removal over a wide pH range (from 2 to 10), and the highest amount of nitrate adsorbed was 25.79 mg g$^{-1}$ in a model nitrate solution. Depending on the aqueous medium (model solutions or real wastewater samples), it was shown that both Langmuir and Freundlich adsorption isotherm models can be used to interpret the adsorption process. It was found that the kinetics are well described by a pseudo-second order model and the nitrate adsorption process can be controlled by chemisorption. The intraparticle diffusion model has been used to identify an adsorption-controlled process by diffusion mechanisms. Adsorption/desorption experiments in column confirmed that MHS could be successfully used in multiple cycles (at least three), indicating the potential of MHS as an alternative to costly commercial adsorbents for the removal of nitrates from wastewater.

**Keywords:** nitrate removal; adsorption; adsorption capacity; column experiment; wastewater

## 1. Introduction

As an essential component of the nitrogen cycle, the nitrate ion is ubiquitous in the environment. Being highly water soluble, it is one of the most widespread contaminants of groundwater and surface water, posing a serious threat to global supplies of drinking water and aquatic ecosystems (as a promoter of eutrophication) [1]. Although it occurs naturally, the increase in nitrate levels in water are especially promoted by the excessive use of nitrogenous fertilizers or manure in intensive agriculture [2–4]. Increased concentrations of nitrate in potable water have been associated with many types of cancer, diabetes, infectious diseases, cyanosis in children, and the possible formation of nitrosamines, which are carcinogenic and can cause baby blue syndrome [4,5]. The WHO has limited the nitrate concentration in potable water to 10 mg L$^{-1}$ (NO$_3^-$−N).

In addition to its high solubility in water, the nitrate ion is also very stable, so its removal could be challenging [6,7]. For this reason, various physical and chemical methods for nitrate removal have been studied and developed, such as adsorption [8], ion exchange [9,10], reverse osmosis [11], electrodialysis [12], denitrification [13], catalytic reduction [14], and many others. Removal of nitrate from water is of utmost importance and therefore optimization of existing technologies is also crucial [15].

The most popular, simple, and efficient methods for nitrate removal are adsorption and ion exchange [3,16]. From an economic perspective, adsorbents should be efficient,

cheap, and highly selective for pollutants. With this in mind, various adsorbents have been tested for the removal of nitrate and other pollutants from water, such as clay, chitosan, zeolite, carbon-based adsorbents, and agro-industrial waste materials (many of which contain lignocellulose) [1]. However, of all the adsorbents listed, activated carbon is still considered the most efficient, probably due to its large surface area and versatility. However, activated carbons available on the market are mainly based on coal, which is a non-renewable resource and often expensive. Therefore, the long-term sustainability of coal-based activated carbon may be in question. To address this issue, more sustainable options are currently being investigated, such as the use of lignocellulosic and other waste materials from the agro-food industry [17]. Lignocellulosic materials are mainly composed of lignin, cellulose, and hemicellulose [18] and can bind a variety of substances due to their structure and chemical composition. These materials have demonstrated great potential for water and wastewater treatment. However, to increase the adsorption capacity or to favorably influence the selectivity of materials to be used as adsorbents, various modifications of their surface are often required. Modification techniques are usually divided into chemical, physical, biological, and electrochemical [4]. Chemical modifications are usually carried out with acids (inorganic and organic), salt and alkali solutions, oxidizing agents, and other chemicals [19].

As a waste material/by-product of the food industry, hazelnut shells (HS) are available in significant quantities in some countries, often not only during the harvest season. The annual production of hazelnuts in 2019/2020 was about 528,070 tons, of which roughly 67% (i.e., 353,897 tons) are shells [20,21]. Hazelnut shells are mostly used as fuel (thermal utilization), since their calorific value is comparable to that of wood. They also have the similar chemical composition, such as wood, HS consists mainly of the lignocellulosic polymers, lignin, cellulose, and hemicellulose. Another possible use of untreated HS based on its chemical composition, is as an adsorbent. Native (unmodified) hazelnut shells have been used for adsorptive removal of copper ions from water [22], dye removal [23] and chlorophenols [24], while modified hazelnut shells (in form of activated carbon or chemically modified) have been found to be effective for lead [25], cadmium, zinc, copper [26], uranium (VI) [27], arsenic (III) [28], chromium (VI) [29], methylene blue [30], crystal violet [31], and taxol (anticancer drug) [32] removal. After adsorptive removal of pollutants, the same material (now loaded with adsorbate) can be used directly as a fuel, eliminating the often difficult step of regeneration and disposal of the used adsorbent [23]. Moreover, HS can also be used for the production of activated carbon after a suitable thermal treatment.

The objective of this research was to prepare a novel adsorbent by chemical modification of hazelnut shells and evaluate its potential for the removal of nitrate from wastewater. The effects of various process parameters on the adsorptive removal of nitrate using the modified HS were investigated in a batch process, namely, initial adsorbent concentration, contact time, initial nitrate concentration, and pH. The regeneration capacity of the novel adsorbent and the possibility of using it in real water treatment systems were tested in fixed bed column experiments

## 2. Materials and Methods

### 2.1. Materials

All chemicals used were of analytical grade. Epichlorohydrin (ECH) and ethylene-diamine were purchased from Sigma Aldrich (Sigma Aldrich, St. Louis, MO, USA), tri-ethylamine Fisher Scientific (Leicestershire, UK) and *N,N*-dimethylformamide (DMF) were from GramMol (GramMol, Zagreb, Croatia). The nitrate solutions were prepared using potassium nitrate (Merck, Darmstadt, Germany). To prepare a 1000 mg $L^{-1}$ (as $N-NO_3^-$) nitrate stock solution, 7.218 g $KNO_3$ was dissolved in 1 L demineralized water. The adsorbate solutions of the desired concentration (10–300 mg $L^{-1}$) were prepared by appropriate dilution of the stock solution.

The model (synthetic) wastewater (MW) was prepared as in Kosjek et al. [33] and the appropriate amount (10–30 mg L$^{-1}$) of KNO$_3$ solution was added. The local confectionery factory (CW) and dairy industry (DW) provided the 24 h composite samples of the real wastewater.

### 2.2. Adsorbent Preparation

PP Orahovica d.o.o., Croatia, kindly provided the hazelnut shells (HS). A laboratory knife mill (MF10 basic, IKA Labortechnik, Germany) equipped with a 1 mm sieve was used to grind the material. The procedure described by Keränen et al. [34], i.e., the epichlorohydrin–triethylamine (ETM) method, was slightly adapted and used for the chemical modification of HS. Here, 2 g HS was mixed with 16 mL DMF and 13 mL ECH at 70 °C. After 45 min, 2.5 mL of ethylenediamine was added to the mixture and stirred for another 45 min at 80 °C. The introduction of amine groups was achieved by adding 13 mL of trimethylamine to the mixture and stirring at 80 °C for 120 min. The obtained modified material was washed with ultra-pure water and dried at 100 °C for 24 h.

### 2.3. Adsorbent Characterization

HS and MHS morphology and surface characteristics were studied using a field emission scanning electron microscope (FE SEM, JOEL, JSM-7000 F, Akishima, Tokyo, Japan). The Perkin Elmer CHNS/O analyzer (II series, Waltham, MA, USA) was used for elemental composition analysis (C, H, N), while the Fourier transform infrared spectrometer (FT-IR) (Cary 630, Agilent Technologies, Santa Clara, CA, USA) was used for the identification of functional groups on the MHS surface involved in nitrate adsorption.

Determination of pH$_{pzc}$ of MHS

The point of zero charge was determined according to Khan and Sarwar [35]. Briefly, in a series of Erlenmeyer flasks (different for each pH of the solution), 0.5 g of MHS reacted with 20 mL of 0.01 M NaCl after adjusting the pH of the solutions (from 2 to 10) with NaOH or HCl. The flasks were shaken for 24 h at 25 °C and 130 rpm. Then, the solutions were filtered and the final pH was measured in each flask, and the difference between the initial and final pH (ΔpH) was calculated. The pH$_{pzc}$ value was determined from the ΔpH versus pH$_{initial}$ plot.

### 2.4. Batch Adsorption Experiments

The adsorption experiments were carried out in a batch-type procedure using a shaker water bath (Bioblock Scientific, Poly-test 20). The adsorption experiments were performed in different aqueous media, namely, a model nitrate solution (MS), a model (synthetic) wastewater (MW), and two real wastewater samples—one from the confectionery industry (CW) and one from the dairy industry (DW).

A description of the batch adsorption experiments is given in Table 1, where $\gamma_{nitrate}$ is the initial nitrate concentration, $\gamma_{adsorbent}$ is the adsorbent concentration, $t$ is the contact time, and $V$ is the volume of the aqueous phase.

**Table 1.** Batch adsorption experiments.

| Experiment | Process Parameters |
| --- | --- |
| Effect of initial MHS concentration | $\gamma_{nitrate}$ = 30 mg L$^{-1}$, $\gamma_{adsorbent}$ = 1−10 g L$^{-1}$, $V$ = 50 mL, pH = native (6.3 (MS), 7.5 (MW), 5.7 (CW), and 9.4 (DW), respectively), $\Theta$ = 25 °C, $t$ = 120 min, $v$ = 130 rpm. |
| Effect of contact time | $\gamma_{nitrate}$ = 30 mg L$^{-1}$, $\gamma_{adsorbent}$ = 4 g L$^{-1}$, $V$ = 50 mL, pH = native (6.3 (MS), 7.5 (MW), 5.7 (CW), and 9.4 (DW), respectively), $\Theta$ = 25 °C, $t$ = 2−1440 min, $v$ = 130 rpm. |
| Effect of initial nitrate concentration | $\gamma_{nitrate}$ = 10−300 mg L$^{-1}$, $\gamma_{adsorbent}$ = 4 g L$^{-1}$, $V$ = 50 mL, pH = native (6.3 (MS), 7.5 (MW), 5.7 (CW), and 9.4 (DW), respectively), $\Theta$ = 25 °C, $t$ = 120 min, $v$ = 130 rpm. |
| Effect of initial solution pH | $\gamma_{nitrate}$ = 30 mg L$^{-1}$, $\gamma_{adsorbent}$ = 4 g L$^{-1}$, $V$ = 50 mL, pH = 2, 4, 6, 7, 8, 10, $\Theta$ = 25 °C, $t$ = 120 min, $v$ = 130 rpm. |

Samples (in Erlenmeyer flasks) were removed from the shaking water bath at predetermined time intervals. After filtration, the residual nitrate concentration was determined spectrophotometrically at 324 nm using a UV/Vis spectrophotometer (Specord 200, Analytic Jena, Germany).

The amount of nitrate removed was expressed as a percentage removal R (%) and calculated as follows:

$$R = \frac{\gamma_0 - \gamma}{\gamma_0} \cdot 100 \tag{1}$$

where $\gamma_0$ is the initial nitrate concentration (mg L$^{-1}$) and $\gamma$ is the nitrate concentration after a predetermined contact time.

The amount of nitrate that was adsorbed onto MHS at equilibrium was calculated as follows:

$$q_e = \frac{(\gamma_0 - \gamma_e)}{m} \cdot V \tag{2}$$

where $q_e$ is the amount of adsorbed nitrate (mg g$^{-1}$), $\gamma_0$ is the initial concentration of nitrate, $\gamma_e$ is the concentration of nitrate at equilibrium, $V$ is the volume of solution (L), and $m$ (g) is the mass of MHS. The effect of temperature on the adsorption process was studied at 25, 35, and 45 °C. Adsorption data were analyzed using the nonlinear form of Langmuir and Freundlich adsorption models, while kinetic data were analyzed using the pseudo-first order, pseudo-second order, and intraparticle diffusion models.

All experiments described above were performed in duplicates and proved to be reproducible.

### 2.5. Column Experiments

The column adsorption experiments were performed using fixed-bed column. The adsorbent MHS (1 g) was packed in a glass column (13 mm × 220 mm). Then, 2 L of a nitrate solution (30 mg L$^{-1}$) was continuously fed to the top of the column using a peristaltic pump (Masterflex L/S, Cole-Palmer Instrum Company, Vernon Hills, IL, USA) at a constant flow rate (10 mL min$^{-1}$) and at 25 °C and natural pH (i.e., the pH was not adjusted and was 6.3 (MS), 7.5 (MW), 5.7 (CW), and 9.4 (DW)). Samples (100 mL) were taken from the bottom of the column to determine nitrate concentration; 200 mL of 0.1 M NaCl and 500 mL of demineralized water were used to regenerate nitrate-loaded MHS (at flow rate of 10 mL min$^{-1}$) in situ. The Equation (3) was used to calculate the saturation capacity ($q_s$, mg g$^{-1}$):

$$q_s = \frac{\gamma_0 V_e - \sum \gamma_n V_n}{m} \tag{3}$$

where $\gamma_0$ is the initial concentration of nitrate (mg L$^{-1}$), $V_0$ is the initial volume of nitrate solution (L), $\gamma_0$ is the concentration of nitrate in fraction n (mg L$^{-1}$), $V_n$ is the volume of fraction n (L), and $m$ is the mass of MHS (g).

## 3. Results and Discussion

### 3.1. Characterization of the Adsorbent

An amount of 2 g of HS was modified and the modification process resulted in 10.6 g of MHS (on a dry weight basis). The characterization of HS and MHS can be found below.

Table 2 shows the results of the elemental analyses of both unmodified and modified HS. It can be seen that the content of nitrogen in MHS (8.14%) is significantly higher than that in HS (0.1%). Ammonium groups were introduced into the material during the modification process, which explains the increase in nitrogen content of MHS compared to HS. This increase is consistent with other research on modified lignocellulosic materials [34,36].

The morphological properties of the surface of the adsorbents were investigated using FESEM imaging. The rougher surface of MHS with more visible cavities compared to HS is shown in Figure 1. Similar results were reported by Yang et al. [37] who observed porous structure in the adsorbent functionalized with triethylamine groups, and by Keränen et al. [34] who used modified bark and peat for nitrate removal.

**Table 2.** Elemental composition of HS and MHS.

| Parameter % Mass | HS | MHS |
|---|---|---|
| C | 48.91 | 45.91 |
| H | 6.28 | 8.76 |
| N | 0.1 | 8.14 |

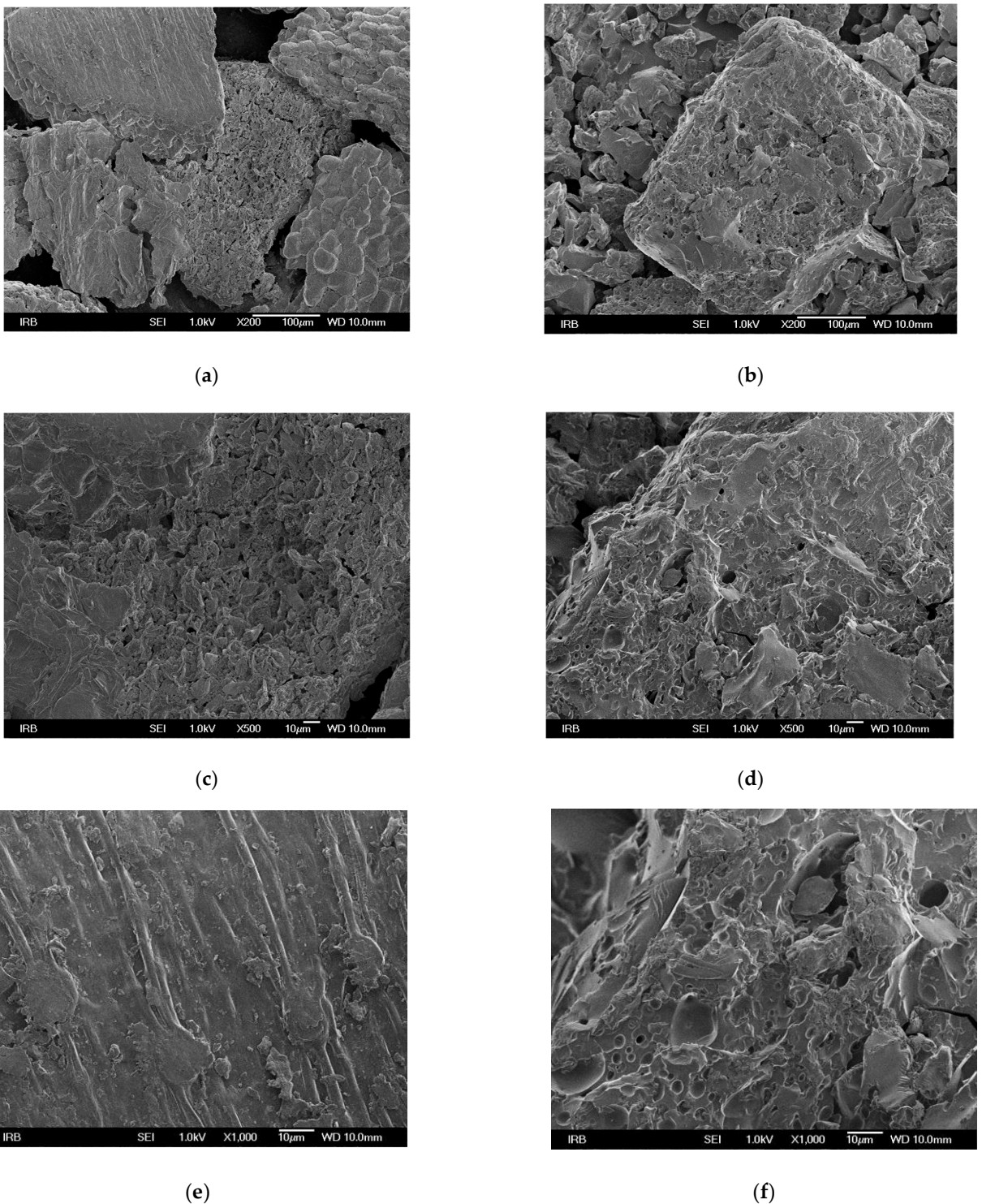

**Figure 1.** FESEM micrographs of HS at different magnifications (**a**) 200, (**c**) 500, (**e**) 1000, and of MHS (**b**) 200, (**d**) 500, (**f**) 1000.

For the identification of the surface functional groups of HS and MHS, FTIR spectroscopy was applied and the respective spectra recorded (Figure 2). The broad band at 3250 cm$^{-1}$ dominates the FTIR spectra of MHS and represents hydroxyl groups (-OH) [37]. The bands can be assigned to the O-H stretching vibrations caused by inter- and intramolecular hydrogen bonding of polymeric compounds (e.g., lignin and cellulose). The peak at 2937 cm$^{-1}$ could be attributed to C-H stretching vibrations of -CH$_2$, while the peak at 2832 cm$^{-1}$ could be attributed to the same vibrations of -CH$_3$ groups [38]. The peaks at 1647 cm$^{-1}$ and 1453 cm$^{-1}$ could be assigned to aromatic cyclic groups and quaternary ammonium groups [3,37], respectively. Similar results were also observed for other quaternized adsorbents [39,40].

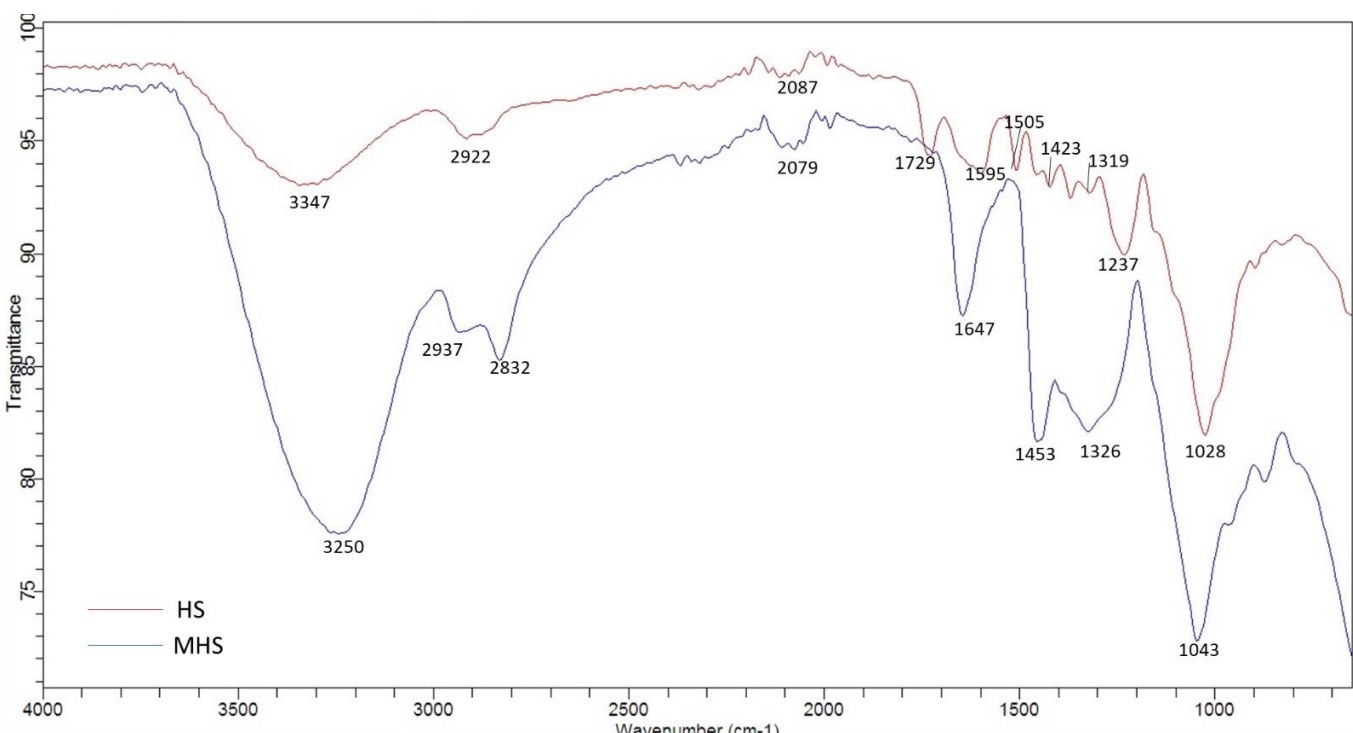

**Figure 2.** FTIR spectra of HS and MHS.

The point of zero charge of the adsorbent (pHpzc) corresponds to a pH at which the surface charge density is equal to zero. Determination of the zero charge point of the adsorbent is important to better understand the electrostatic interactions between the surface of the adsorbent and the adsorbate at a given pH [41,42]. From the results presented in Figure 3, it can be seen that the pH$_{pzc}$ of MHS is 5.6. At pH values below pH$_{pzc}$, the surface of the adsorbent is charged positively, which is conducive to the adsorption of anions. If pH > pH$_{pzc}$, the surface of the adsorbent is charged negatively, which is conducive to the adsorption of cations [42].

### 3.2. Batch Adsorption Studies

### 3.2.1. Effect of Contact Time

The removal of nitrate from the four aqueous media considered with time is shown in Figure 4. It was found that the adsorption efficiency of nitrate ions gradually increased with increasing contact time. Within the first 15 min, about 65%, 42%, 37%, and 30% of nitrate was removed from MS, MW, CW, and DW, respectively, while the adsorption capacities were 6.75, 4.44, 3.8, and 3.5 mg g$^{-1}$. Equilibrium was reached after 60 min. Similar results were reported by Keränen et al. [34] using modified pine sawdust and [43] who studied the adsorption of nitrate onto quaternary starch derivatives. In addition, Stjepanović et al. [44] reported that ETM-modified grape seeds could be considered as an

alternative for the commercial ion exchanger Relite A490. The lower percent removal from wastewater samples is to be expected and could be explained by the fact that such a complex matrix contains other ions, which prevent nitrates from adsorbing by competing for the same adsorption sites.

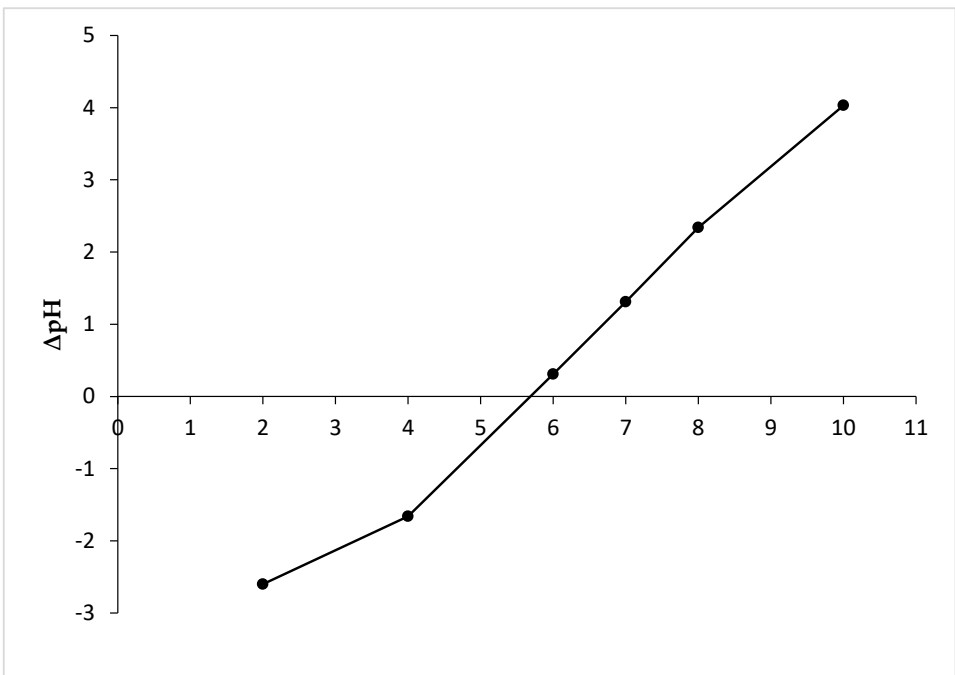

**Figure 3.** $pH_{pzc}$ of MHS.

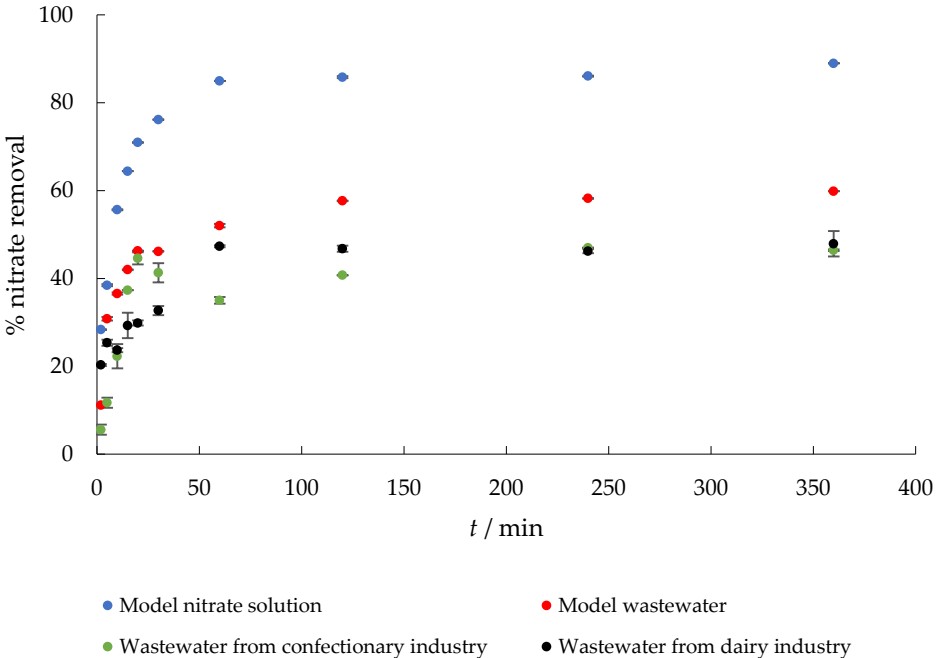

**Figure 4.** The effect of contact time on the adsorption of nitrate to MHS ($\gamma_{nitrate}$ = 30 mg L$^{-1}$, $\gamma_{adsorbent}$ = 4 g L$^{-1}$, pH = 6.3 (MS), 7.5 (MW), 5.7 (CW), 9.4 (DW), $\Theta$ = 25 °C, $v$ = 130 rpm).

In discussing the adsorption mechanism, electrostatic interaction and coulombic forces between the quaternary ammonium functional group of MHS and $NO_3^-$ are the most probable mechanisms of nitrate adsorption on MHS [45]. During adsorption, nitrate ions were briefly exchanged by Cl$^-$ ions. Therefore, ion exchange can be considered as the main

mechanism involved in the adsorption of nitrate ions onto the MHS. This is supported by other work [46], which determined the concentration of chlorine after adsorption, and found that it may be ion exchange due to the increased chlorine concentration.

### 3.2.2. Effect of Adsorbent Concentration

In general, more adsorbate is removed from a solution when a higher concentration of adsorbent is used because more active sites are available and the surface area is increased [42]. The amount of nitrate adsorbed onto MHS by using various adsorbent concentration is shown in Figure 5. With the increase in the adsorbent concentration from 1 to 10 g $L^{-1}$ the amount of adsorbed nitrate also increased.

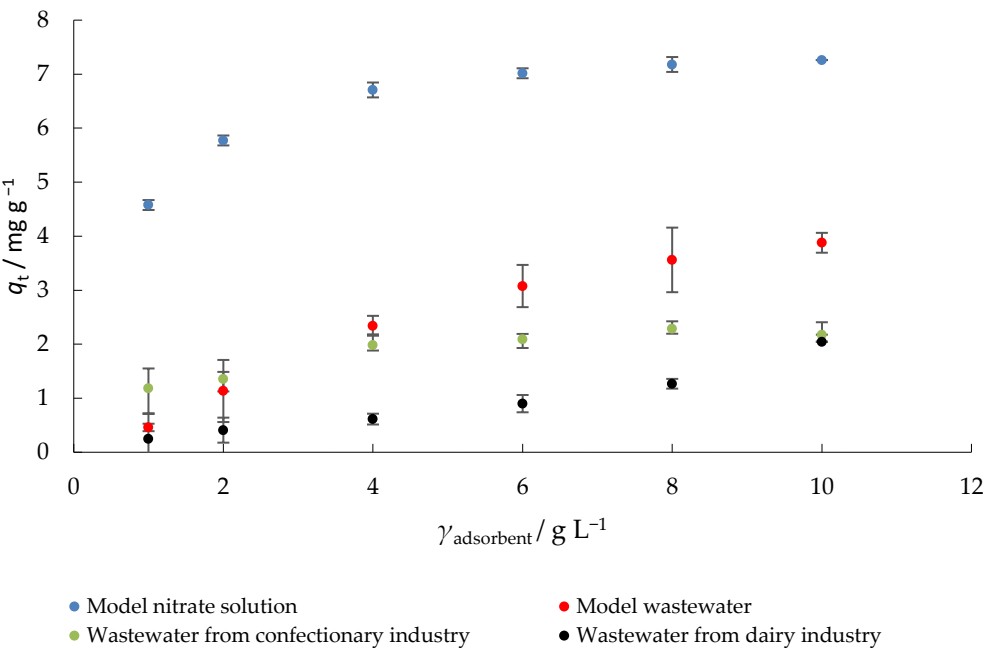

- Model nitrate solution
- Wastewater from confectionary industry
- Model wastewater
- Wastewater from dairy industry

**Figure 5.** The effect of adsorbent concentration on the adsorption of nitrate to MHS ($\gamma_{nitrate}$ = 30 mg $L^{-1}$, $t$ = 120 min, pH = 6.3 (MS), 7.5 (MW), 5.7 (CW), 9.4 (DW), $\Theta$ = 25 °C, $v$ = 130 rpm).

Using a model solution of nitrates, the maximum removal efficiency (percent nitrate removal) and amount of nitrate adsorbed on MHS (93%, 7.26 mg $g^{-1}$) were obtained at the highest adsorbent concentration used. The highest efficiency of nitrate removal in the model wastewater was 54%, in the wastewater from confectionery industry 25%, and in the wastewater from dairy industry 27%, while the amounts of nitrate adsorbed onto MHS were 3.9, 2.2, and 2.1 mg $g^{-1}$, respectively, under the same experimental conditions (10 g $L^{-1}$ adsorbent concentration). When model nitrate solutions were tested, increasing the adsorbent mass had no significant effect on the percent nitrate removal. To keep operating costs as low as possible, the concentration of adsorbent chosen for further experiments was 4 g $L^{-1}$. Moreover, at the adsorbent concentration of 4 g $L^{-1}$, a good balance was observed between the percent nitrate removal and the amount of nitrate adsorbed onto MHS. Similar trends have been observed in other studies, e.g., Kalaruban et al. [47] reported higher nitrate removal efficiency when the concentration of the adsorbent (corncob and coconut copra modified by amine grafting) increased. Hafshejani et al. [42] also observed an increase in nitrate removal with the increase in adsorbent dosage in modified sugarcane bagasse.

### 3.2.3. Effect of Initial Nitrate Concentration

The results of nitrate adsorption by MHS at different initial concentrations are presented in Figure 6. The highest increase in percent nitrate removal was observed when a model nitrate solution was used, which was expected since there is no competition between

different ions. Nitrate concentration on MHS increased from 2.5 ($\gamma_{nitrate}$ = 10 mg L$^{-1}$) to 25.8 mg g$^{-1}$ ($\gamma_0$ = 300 mg L$^{-1}$) in model nitrate solution. In MW, $q_t$ increased from 0.8 to 11.6 mg g$^{-1}$, in CW from 1.4 to 21.5 mg g$^{-1}$ and in DW from 2.1 to 13.7 mg g$^{-1}$ under the same experimental conditions. When the concentration gradient (as a driving force) increases as a result of the increase in adsorbate concentration, the adsorption of nitrate on MHS also increases [48].

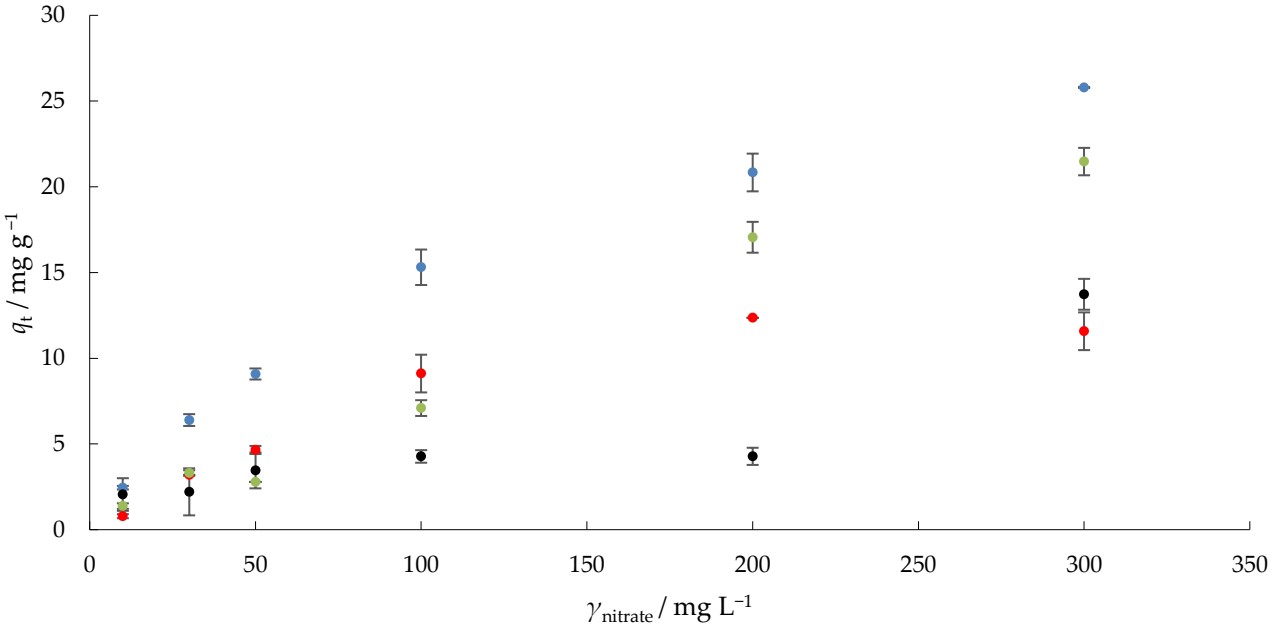

● Model nitrate solution　● Model wastewater　● Wastewater from confectionary industry　● Wastewater from dairy industry

**Figure 6.** The effect of initial nitrate concentration on the adsorption of nitrate to MHS ($\gamma_{adsorbent}$ = 4 g L$^{-1}$, $t$ = 120 min, pH = 6.3 (MS), 7.5 (MW), 5.7 (CW), 9.4 (DW), $\Theta$ = 25 °C, $v$ = 130 rpm).

### 3.2.4. Effect of pH

The effect of initial pH on adsorptive nitrate removal from MS, MW, CW, and DW is shown in Figure 7. In a pH range from 4 to 10, the maximum percentage of nitrate removal was achieved, while this percentage slightly decreased at pH 2. Nitrate removal from MW and CW showed a decreasing trend from pH 2 to 10, which can be attributed to the OH$^-$ ions that compete for the same adsorption sites. When looking at nitrate removal from DW, it appears that it was not significantly affected by pH; adsorption of nitrates on MHS was almost constant over time. The only exception was observed at pH 8, when a slight increase in the amount of nitrate removed. Similar trends were reported by Banu and Meenakshi [39], who studied the pH effect of pH over a wide range from 2 to 11 on the nitrate removal using quaternized melamine–formaldehyde resin, and Sowmya and Meenakshi [49], who used quaternized chitosan beads, and also Chauhan et al. [43] investigating adsorption properties of quaternary ammonium-functionalized starch.

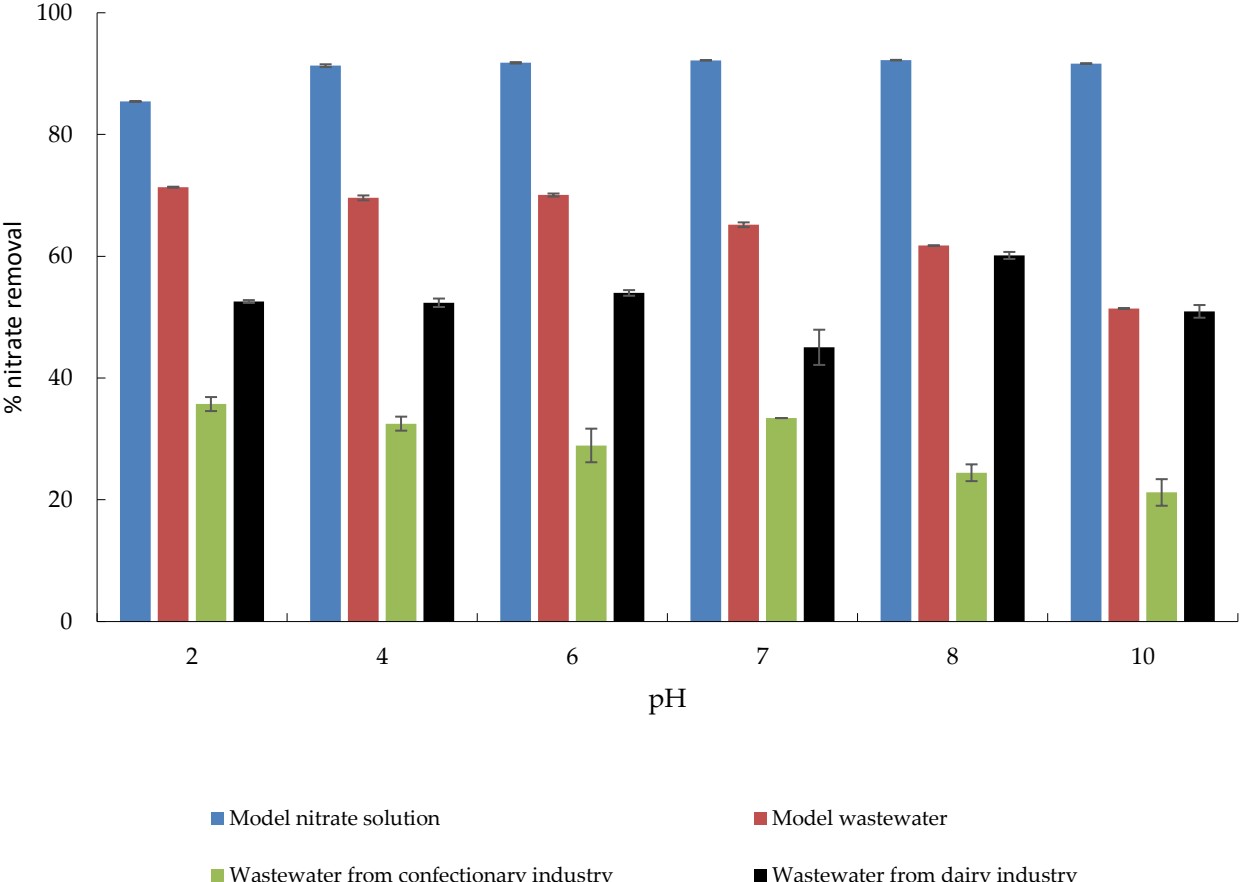

**Figure 7.** The effect of pH on the adsorption of nitrate to MHS ($\gamma_{nitrate}$ = 30 mg L$^{-1}$, $\gamma_{adsorbent}$ = 4 g L$^{-1}$, $t$ = 120 min, $\Theta$ = 25 °C, $v$ = 130 rpm).

*3.3. Adsorption Isotherms*

Non-linearized Langmuir and Freundlich models were used for analyses of the obtained adsorption data. The Langmuir model theory explains that adsorption takes place at specific sites on the adsorbent that are homogeneous and is described as follows [50]:

$$q_e = \frac{q_m \cdot K_L \cdot \gamma_e}{1 + K_L \cdot \gamma_e} \tag{4}$$

where $\gamma_e$ (mg L$^{-1}$) is the concentration of nitrate at equilibrium, $q_e$ (mg g$^{-1}$) is the amount of nitrate adsorbed per mass of adsorbent, $q_m$ (mg g$^{-1}$) is the maximum amount of nitrate adsorbed, and $K_L$ (L mg$^{-1}$) is the Langmuir constant. To describe whether the adsorption of nitrate on MHS is a process that is unfavourable ($R_L > 1$), favourable ($0 < R_L < 1$), linear ($R_L = 1$), or irreversible ($R_L = 0$), the dimensionless constant $R_L$ (equilibrium parameter) was calculated as follows: [50]:

$$R_L = \frac{1}{1 + K_L \cdot \gamma_o} \tag{5}$$

where $\gamma_0$ (mg L$^{-1}$) is the highest initial concentration of nitrate. The $R_L$ values given in Table 3 are 0.061, 0.164, 0.959, and 0.011 (for MS, MW, CW, and DW, respectively), indicating that adsorptive removal of nitrate using MHS as adsorbent was a favourable process under the experimental conditions applied.

**Table 3.** Isotherm parameters for the nitrate adsorption onto MHS from different aqueous solutions.

| Isotherm Model | MS | MW | CW | DW |
|---|---|---|---|---|
| $q_{m\,exp.}/\text{mg g}^{-1}$ | 25.79 | 11.57 | 21.47 | 13.73 |
| **Langmuir** | | | | |
| $q_{m\,cal.}/\text{mg g}^{-1}$ | 26.508 | 16.031 | 756.6 | 66.204 |
| $K_L/\text{L mg}^{-1}$ | 0.051 | 0.017 | $1.4\cdot10^{-4}$ | 0.301 |
| $R_L$ | 0.061 | 0.164 | 0.959 | 0.011 |
| $R^2$ | 0.961 | 0.950 | 0.968 | 0.878 |
| $MSE$ | 2.617 | 0.935 | 1.865 | 49.595 |
| $RMSE$ | 1.618 | 0.967 | 1.366 | 7.042 |
| **Freundlich** | | | | |
| $K_F/(\text{mg g}^{-1}\,(\text{L/mg})^{1/n})$ | 4.081 | 1.044 | 0.099 | 23.154 |
| $n$ | 2.803 | 2.141 | 0.993 | 4.301 |
| $R^2$ | 0.995 | 0.862 | 0.968 | 0.958 |
| $MSE$ | 0.312 | 2.587 | 1.871 | 17.005 |
| $RMSE$ | 0.558 | 1.609 | 1.368 | 4.124 |

The Freundlich adsorption isotherm model is described by the Equation (6) and assumes the adsorption in a multimolecular layer in which interactions occur between adsorbate particles [48]:

$$q_e = K_f \gamma_e^{\frac{1}{n}} \tag{6}$$

where $q_e$ (mg g$^{-1}$) is the amount of nitrate ions adsorbed onto MHS at equilibrium, $\gamma_e$ (mg L$^{-1}$) is the nitrate concentration at equilibrium, $K_F$ is the constant indicating the adsorption capacity of the adsorbent, and $n$ is the constant indicating whether the adsorption is a chemical process ($n < 1$), favourable physical process ($n > 1$), or linear ($n = 1$). The constants $n$ in this study were 2.803, 2.141, and 4.301 for MS, MW, and DW, respectively, indicating a favourable physical process, while for CW the $n$ value is 0.993, showing that it is a chemical process.

The orientation of the curves in Figure 8 indicates an "L" type, subgroup 2, for MS, MW, and DW; and a "C" type, subgroup 1, for CW, according to the isotherms classification by Giles [51], which are the classical and best-known Langmuir isotherms.

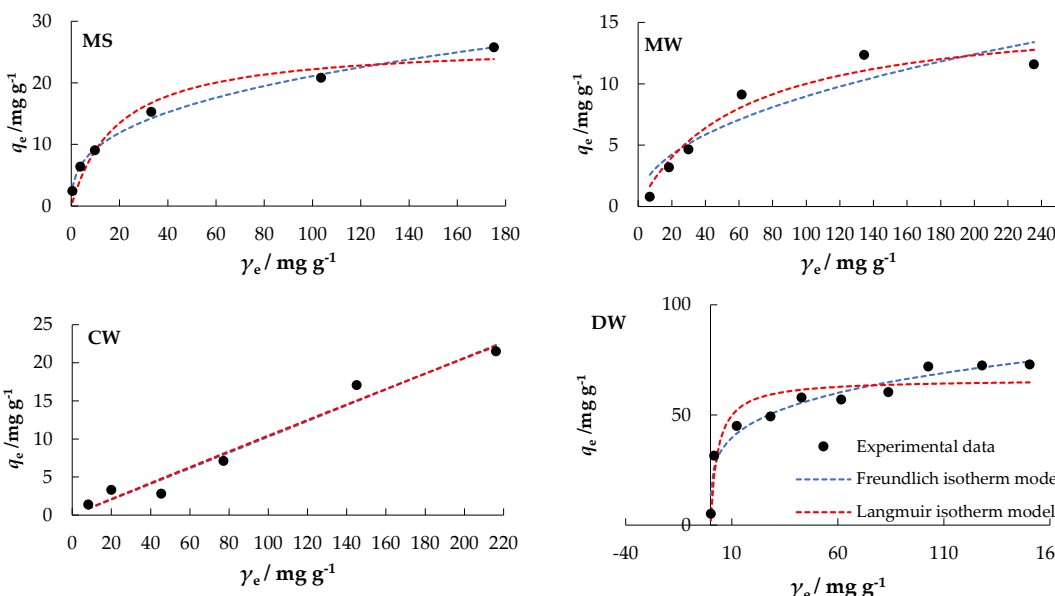

**Figure 8.** Freundlich and Langmuir isotherms of nitrate adsorption onto MHS at $\Theta = 25\ ^\circ$C.

In a comparative analysis of the values of the regression coefficients, $R^2$, MSE, and RMSE, the experimental data (Table 3) best fit the Freundlich model for MS and DW, while the Langmuir model fit best for MW. Both, the Langmuir and Freundlich model could be used to interpret CW data.

The obtained adsorption capacity of MHS for nitrate (Langmuir model) was compared with adsorption capacity values reported in the literature for other adsorbents (Table 4). The adsorption capacity obtained in this study is in accordance with the adsorption capacities reported in literature.

**Table 4.** Adsorption capacity for nitrate of various adsorbents.

| Adsorbent | Adsorption Capacity $q_{m\ Langmuir}$/mg g$^{-1}$ | References |
|---|---|---|
| Quaternized melamiformaldehydeyde resin | 124.16 | [39] |
| Quaternized resin with acrylonitrile/divinylbenzene/ vinyl benzyl chloride | 59.7 | [52] |
| Quaternized pine bark | 46.9 | [53] |
| Quaternized pine sawdust | 29.5 | [54] |
| Greenish clay | 27.77 | [16] |
| Spent mushroom compost treated with iron(III) chloride hexahydrate | 19.88 | [55] |
| Chitin | $2 \cdot 10^{-4}$ | [56] |
| Triethylamine-functionalized polystyrene microsphere | 47.27 | [37] |
| Modified brewers' spent grain | 24.16 | [57] |
| Modified grape seeds | 27.47 | [44] |
| Modified hazelnut shells | 26.51 | This study |

*3.4. Thermodynamic Study*

The effect of temperature on the adsorption of nitrate to MHS was studied at three different temperatures (298.15, 308.15, and 318.15 K) by varying the initial nitrate concentrations from 10 to 300 mg/L. Increasing the temperature in MS had no significant effect on nitrate removal, while in MW, CW, and MW increasing the temperature showed a slight increase in the percent nitrate removal (31–45% in MW, 40–42% in CW, and 60–61% in DW), when the solution temperature increased from 298.15 to 318.15 K, indicating an endothermic adsorption process. The thermodynamic parameters: free Gibbs energy ($\Delta G$), standard entropy change ($\Delta S$), and standard enthalpy change ($\Delta H$) were calculated to show the adsorption mechanism and the feasibility of adsorption using the van't Hoff plot ($\ln K_c$ vs, $1/T$) and equation [34]:

$$\ln K_c = \frac{\Delta S}{R} - \frac{\Delta H}{RT} \tag{7}$$

where $K_c$ is the thermodynamic equilibrium constant, $R$ is the universal gas constant (8.314 j/mol K), and $T$ is the absolute temperature (K).

All calculated thermodynamic parameters are given in Table 5.

**Model nitrate solution**. The spontaneity of the adsorption process of nitrate on MHS was indicated by the $\Delta G$ values, which were negative. The endothermic nature of the process was confirmed by a positive $\Delta H$, while a positive $\Delta S$ indicated disorder in the system caused by nitrate adsorption. The adsorption process was more favorable at high temperatures, as confirmed by the fact that the $\Delta G$ increased with the increase in temperature. This may be associated with electrostatic interaction and ion exchange [58].

**Model wastewater and wastewater from confectionery industry**. Positive $\Delta G$ values in MW and CW showed that the adsorption process of nitrate was not spontaneous. Negative $\Delta H$ values indicate an exothermic adsorption process [59] and together with negative $\Delta S$ values suggest that the process is spontaneous at very low temperatures.

**Wastewater from dairy industry**. In this case, the positive $\Delta G$ and $\Delta H$ values and the negative $\Delta S$ values suggest that the adsorption process is not spontaneous at all.

**Table 5.** Thermodynamic parameters for nitrate removal by MHS.

| | $T$/K | $K_c$ | $\Delta G$/kJ mol$^{-1}$ | $\Delta H$/kJ mol$^{-1}$ | $\Delta S$/J mol$^{-1}$K$^{-1}$ |
|---|---|---|---|---|---|
| | 298.15 | 3.145 | −2.840 | | |
| MS | 308.15 | 5.433 | −4.196 | 20.328 | 78.468 |
| | 318.15 | 5.234 | −4.103 | | |
| | 298.15 | 0.309 | 2.912 | | |
| MW | 308.15 | 0.139 | 4.896 | −26.332 | −99.319 |
| | 318.15 | 0.160 | 4.542 | | |
| | 298.15 | 0.145 | 4.786 | | |
| CW | 308.15 | 0.181 | 4.240 | −2.104 | −22.436 |
| | 318.15 | 0.137 | 4.932 | | |
| | 298.15 | 0.162 | 4.510 | | |
| DW | 308.15 | 0.165 | 4.467 | 2.044 | −8.295 |
| | 318.15 | 0.171 | 4.381 | | |

### 3.5. Adsorption Kinetics

Adsorption kinetics for nitrate removal were also studied to investigate the mechanisms (chemical reaction or particle/film diffusion) engaged in the adsorption of nitrate on MHS and to determine the critical step of adsorption rate. The kinetic data were analyzed by pseudo-first and pseudo-second order kinetic models and the Weber and Morris intraparticle diffusion model. Tables 6 and 7 show the kinetic parameters of the above models, which are calculated from the slopes and intercepts of the corresponding plots. The correlation coefficients of the linear regression were used as the basis for evaluating the applicability of model to the experimental data ($R^2$).

**Table 6.** Parameters of the pseudo-first-order and pseudo-second-order kinetic models for the removal of nitrate by MHS at $\Theta = 25\ ^{\circ}$C.

| Kinetic Model | MS | MW | CW | DW |
|---|---|---|---|---|
| $q_{\mathrm{m\ exp.}}$/mg g$^{-1}$ | 6.748 | 4.439 | 3.803 | 3.495 |
| **Pseudo-first order** | | | | |
| $q_{\mathrm{m\ 1}}$/mg g$^{-1}$ | 2.077 | 1.711 | 1.560 | 1.190 |
| $k_{1}$/min$^{-1}$ | 0.009 | 0.011 | 0.005 | 0.010 |
| $R^2$ | 0.776 | 0.910 | 0.506 | 0.675 |
| **Pseudo-second order** | | | | |
| $q_{\mathrm{m\ 2}}$/mg g$^{-1}$ | 6.775 | 4.462 | 3.830 | 3.516 |
| $k_{2}$/g mg$^{-1}$ min$^{-1}$ | 0.023 | 0.031 | 0.013 | 0.035 |
| $R^2$ | 1 | 1 | 0.999 | 0.999 |

**Table 7.** Parameters of the Weber and Morris intraparticle diffusion model for the removal of nitrate by MHS ($\gamma_{nitrate}$ = 30 mg L$^{-1}$, $\gamma_{adsorbent}$ = 4 g L$^{-1}$, $t$ = 2–1440 min, pH = 6.3 (MS), 7.5 (MW), 5.7 (CW), 9.4 (DW), $\Theta$ = 25 °C, $v$ = 130 rpm).

| Intraparticle Diffusion Model | MS | MW | CW | DW |
|---|---|---|---|---|
| $k_{i1}$/mg g$^{-1}$ min$^{-0.5}$ | 0.932 | 0.603 | 0.968 | 0.210 |
| $C_1$ | 0.961 | 0.540 | <0 | 1.217 |
| $R_1^2$ | 0.965 | 0.833 | 0.972 | 0.890 |
| $k_{i2}$/mg g$^{-1}$ min$^{-0.5}$ | 0.013 | 0.015 | 0.032 | 0.004 |
| $C_2$ | 6.289 | 3.974 | 2.680 | 3.345 |
| $R_2^2$ | 0.824 | 0.505 | 0.739 | 0.501 |

Table 6 shows that the $q_{m\,exp.}$ values are not in agreement with the calculated $q_{m\,1}$ values, indicating that the pseudo-first-order model is not well suited for modelling the kinetic data. The coefficient $R$, and the calculated $q_{m\,2}$ values are closer to the experimental values, suggesting that the adsorption kinetics is better described by the pseudo-second-order kinetic model and the nitrate adsorption process could be controlled by chemical adsorption. However, the $q$ vs. ln $t$ plot does not pass through the axis origin (data not shown); although it is linear, supporting the conclusion that the adsorption rate could be controlled by mass transport either in the liquid phase or within the particles. Therefore, to clarify the diffusion mechanism, the diffusion model of Weber and Morris was used for further analysis of the kinetic data. Two linear steps can be seen at the intraparticle diffusion model plot, indicating multilinearity (Figure 9).

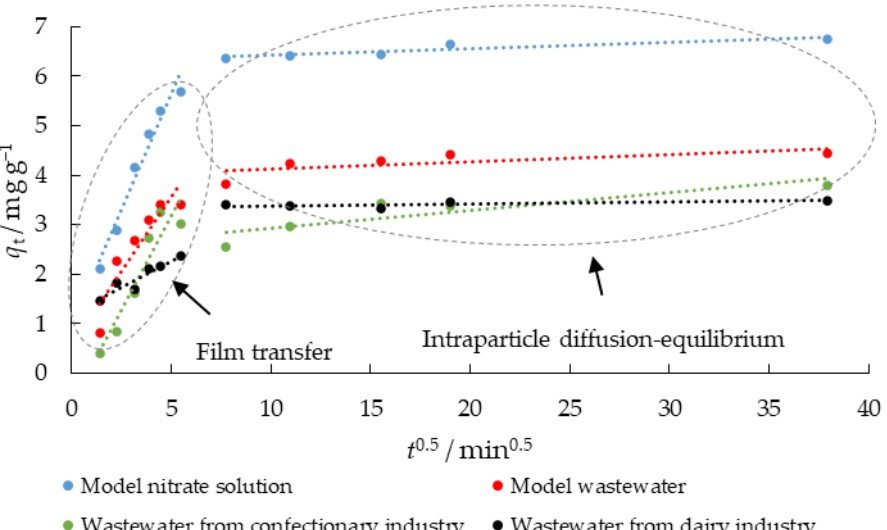

**Figure 9.** Intraparticle diffusion plots for nitrate removal using MHS ($\gamma_{nitrate}$ = 30 mg L$^{-1}$, $\gamma_{adsorbent}$ = 4 g L$^{-1}$, $t$ = 2–1440 min, pH = 6.3 (MS), 7.5 (MW), 5.7 (CW), 9.4 (DW), $\Theta$ = 25 °C, $v$ = 130 rpm).

According to Weber Jr. [60], the first step here could be film transport or film diffusion (external mass transfer), and the second step is associated with intraparticle diffusion, where the direction of nitrate diffusion is from the outside of the adsorbent into the pores. Table 7 shows the modelled diffusion parameters ($k_i$ and $C$) for nitrate removal using MHS. The values for $k_{i1}$ were 0.932, 0.603, 0.968, and 0.210 mg g$^{-1}$ min$^{-0.5}$ in MS, MW, CW, and DW, respectively. The intraparticle diffusion with $k_{i2}$ 0.013, 0.015, 0.032, and 0.004, while $C_2$ was 6.289, 3.974, 2.680, and 3.345 when MS, MW, CW, and DW were used as aqueous media, respectively, was a second phase.

### 3.6. Breakthrough and Desorption Studies

The fixed-bed mode is a reliable method to further test the usability and practicality of a particular adsorbent [37]. The obtained breakthrough curves are shown in Figure 10.

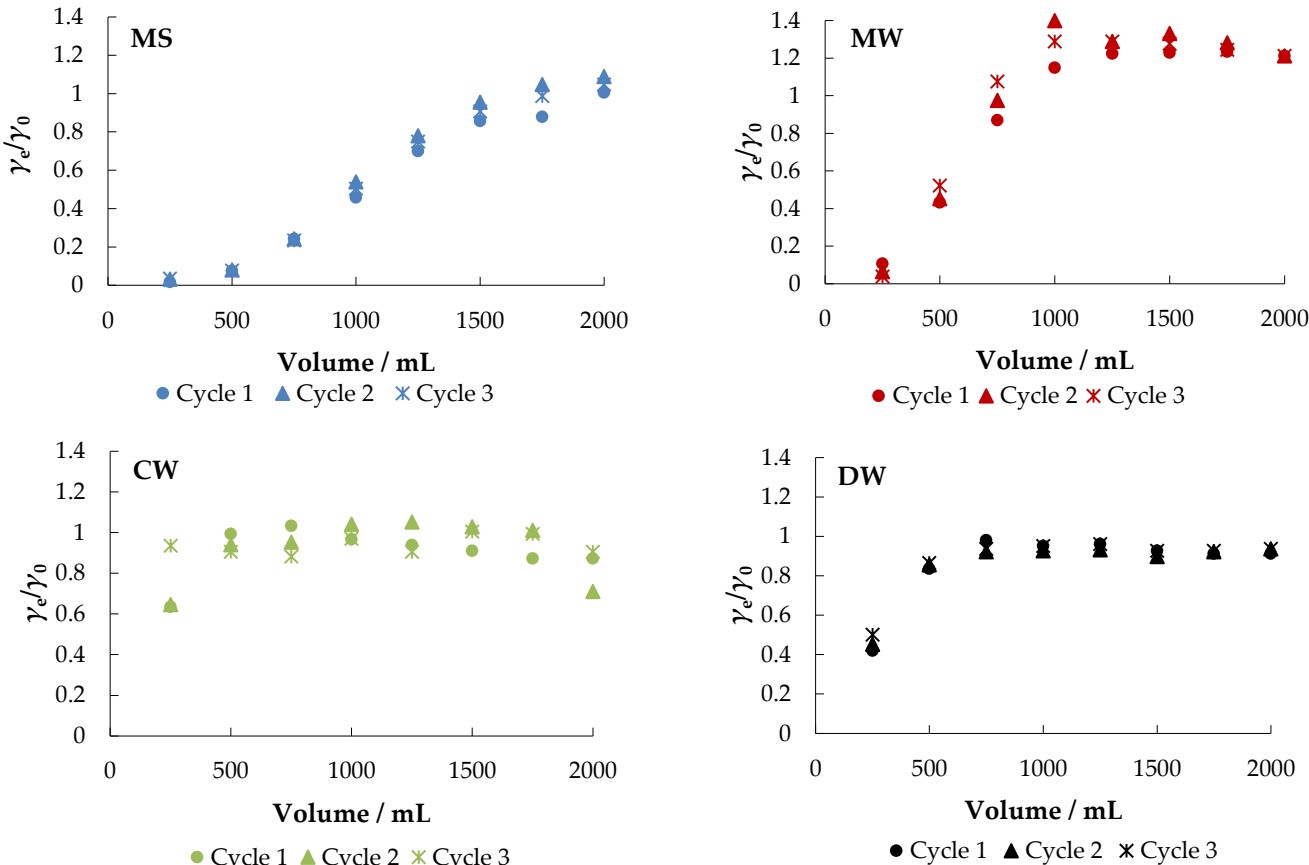

**Figure 10.** Breakthrough curves for the adsorption of nitrate ions from different aqueous solutions onto MHS (bed volume 4 mL, $\gamma_{nitrate}$ = 30 mg L$^{-1}$).

Figure 10 shows that MHS can be used (regenerated) and reused in multiple cycles (at least three cycles) after regeneration.

After the first cycle in MS, about 98% of the nitrate was removed. When the adsorbent was exhausted, regeneration was performed with 0.1 M NaCl at a flow rate of 10 mL min$^{-1}$ and then the system was washed with demineralized water. A negligible decrease in saturation adsorption capacity was noticed between the first (30.04 mg g$^{-1}$), second (27.88 mg g$^{-1}$), and third cycles (26.27 mg g$^{-1}$), indicating that the adsorbent remained stable and, more importantly, reusable.

As expected, faster breakthrough was observed when MW, CW, and DW were used as other ions present competed for the adsorption sites. The percent removal in MW was about 89%, but the saturation capacity after the first cycle was much lower than in MS (4.15 mg g$^{-1}$). The results of the desorption experiment in CW and DW are similar: the breakthrough is very fast with saturation capacities of 5.73 mg g$^{-1}$ and 9.29 mg g$^{-1}$, respectively. The obtained results are supported by the results reported in other studies [53,57].

### 4. Conclusions

This study presented a possibility of modifying waste material from the agri-food industry (i.e., hazelnut shells) that could serve as an alternative, low-cost adsorbent for the removal of nitrate from water and wastewater. Adsorption of nitrate ions on modified hazelnut shells (MHS) was rapid and the equilibrium was reached within 60 min in all aqueous solutions tested. Nitrate adsorption was efficient over a wide pH range



(from 4 to 10). Both Langmuir and Freundlich adsorption isotherm models can be used to interpret the adsorption process. A better fit of the kinetic data was achieved when the pseudo-second-order model was used. The intraparticle diffusion model suggested two steps during the adsorption process: film diffusion and intraparticle diffusion. Finally, the column experiments showed that the MHS could be successfully used in multiple cycles.

**Author Contributions:** Conceptualization, methodology, validation, formal analysis, investigation, and writing—original draft preparation M.S.; writing—review and editing, N.V. and M.H.-S.; supervision, M.H.-S. All authors have read and agreed to the published version of the manuscript.

**Funding:** This research was funded by the Faculty of Food Technology Osijek, Croatia.

**Institutional Review Board Statement:** Not applicable.

**Informed Consent Statement:** Not applicable.

**Data Availability Statement:** Data Sharing is not applicable.

**Acknowledgments:** Financial support from the Faculty of Food Technology Osijek, Croatia is gratefully acknowledged. Authors wish to thank Mirela Kopjar, Faculty of Food Technology Osijek for FTIR spectra recordings using the equipment funded by the project HRZZ- UIP-2013-11-6949.

**Conflicts of Interest:** The authors declare no conflict of interest.

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
