# Peer review of "Modified Hazelnut Shells as a Novel Adsorbent for the Removal of Nitrate from Wastewater"

_water, doi:10.3390/w14050816_

Round 1

Reviewer 1 Report

Reviewers Comments

Manuscript ID:  water-1587028

Title Modified Hazelnut Shells as a Novel Adsorbent for the Removal of Nitrate from Wastewater

Journal: Water

General comments

The study reports the use and evaluation of chemical modification hazelnut shells for the removal of nitrate from water and wastewater. The manuscript is quite interesting, but overall similarity index of 45% and 16% from single source jeopardizes its novelty. Though the manuscript is well written and presents significant vital experimental results analysis, yet it lacks significant aspects related characterizations, nitrate removal adsorption mechanisms presentation and interpretations. A major revision is needed prior to the acceptance of manuscript as per the following comments provided below 

Specific comments

  1. The maximum adsorption capacity and major nitrate sorption mechanisms are missing in the abstract.
  2. Line 14; avoid adding general and fundamental info in abstract
  3. Introduction part not sufficient information is provided. A more comprehensive review on the subject matter with relevant and most updated references is needed.
  4. Adsorption equilibrium and thermodynamic studies are missing. Adsorption mechanism cannot be understood without equilibrium and thermodynamic data. As such authors should add equilibrium and thermodynamic studies to complement present data, discussions, and mechanisms elucidation to help fully understanding the potentials of the new adsorbent in pollution
  5. Correct mechanisms for the metals adsorption should be properly clarified and established using the full characterization of the final adsorbent using techniques such as BET, SEM, EDX, FTIR, TGA, and XRD, elemental compositions analyses and surface charges (zeta potential, point of zero charge) results for both before and after adsorption (each one superimposed)
  6. Provide and discuss the detailed and comparative for the precursor materials and the final adsorbent before and after adsorption to further support understanding and interpretation of the adsorption mechanism.
  7. equilibrium and thermodynamics models’ fittings parameters; R2, models error and well as RMSE should be provided and serve as a guide for selecting the best model for better mechanisms interpretations

Author Response

The authors would like to thank the reviewer 1 for the kind assessment of our manuscript. Please find enclosed the point-by-point replies to the comments regarding the manuscript Water-1587028.

Reviewer 2 Report

This study investigated the removal efficiency of nitrate from wastewater by modified hazelnut shell as a novel adsorbent. The effects of different operation factors were studied and the results are interesting. Specific comments can be found below.

  1. Line 62-72, the application of hazelnut shell in adsorption should be added in this paragraph.
  2. Line 100 and 101, “DMF” and “ECH” have been defined in Line 84 and 82 already.
  3. Line 99-105, it is was abnormal that the modified hazelnut shell was dried at 100 oC, which was prepared below100 o
  4. Line 146-147, “at a natural solution pH”, what was exactly the pH value?
  5. Line 173-178, relevant literature should be cited here.
  6. Figure 2, the name of the y coordinate can not be seen clearly.
  7. Line 200-202, have the authors measured the zeta potential of the modified hazelnut shell to give solid evidence?
  8. Error bars are missing in Figure 3.
  9. Use uniform format for “mg/L” and “mg L-1” throughout the whole manuscript.

Author Response

The authors would like to thank the reviewer 2 for the kind assessment of our manuscript. Please find enclosed the point-by-point replies to the comments regarding the manuscript Water-1587028.

Reviewer 3 Report

This work has presented modified hazelnut shells as a possible, low-cost adsorbent for the removal of nitrate from water and wastewater. This paper is organized and nicely written. After reviewing the paper, I think this manuscript can be under consideration of acceptance with a minor revision.

In figure 6. make the label some distance away from the axis for MS and CW and MW samples.

Author Response

The authors would like to thank the reviewer 3 for the kind assessment of our manuscript. Please find enclosed the point-by-point replies to the comments regarding the manuscript Water-1587028.

Reviewer 4 Report

The article Modified Hazelnut Shells as a Novel Adsorbent for the Removal of Nitrate from Wastewater is interesting and well written. The characterization of the novel adsorbent, modified hazelnut shell, was performed. The adsorbent was found to be effective in nitrate removal over a wide pH range and can be a promising alternative for the removal of nitrates from wastewater.

I would recommend minor corrections:

- Figure 1 should be further explained;

-In line 192, the numbers 6.75, 4.44, 3.8, and 3.5 are missing units of measure;

-On the ninth and tenth pages, the two figures are numbered “Figure 6”;

-The research methodology is described in the results section, lines 157-159, 184-186, 351-354. The methodology should be described in Chapter 2.

Author Response

The authors would like to thank the reviewer 4 for the kind assessment of our manuscript. Please find enclosed the point-by-point replies to the comments regarding the manuscript Water-1587028.

Round 2

Reviewer 1 Report

Reviewers Comments

Manuscript ID:  water-1587028

Title Modified Hazelnut Shells as a Novel Adsorbent for the Removal of Nitrate from Wastewater

Journal: Water

Comments

  • The authors have tried to review the manuscript by addressing some of the reviewers’ comments. However, the failed to overall similarity index which is until now at an unacceptable level of 42% and up to a 13 % from single source (from the original 45% and 16%) which too high and unacceptable. This questioned the originality and merit of the manuscript as well as the work The authors should abid by the journal requirement which if not, the article should be rejected

  • Batch adsorption experiments design under section 2.4 should be provided in Tabular for clarity

Author Response

The authors would like to thank the reviewer 1 for the kind assessment of our manuscript. Please find enclosed the reply to the comment regarding the manuscript Water-1587028.
